# Exaptation of Retroviral Syncytin for Development of Syncytialized Placenta, Its Limited Homology to the SARS-CoV-2 Spike Protein and Arguments against Disturbing Narrative in the Context of COVID-19 Vaccination

**DOI:** 10.3390/biology10030238

**Published:** 2021-03-19

**Authors:** Malgorzata Kloc, Ahmed Uosef, Jacek Z. Kubiak, Rafik M. Ghobrial

**Affiliations:** 1Houston Methodist Research Institute, Houston, TX 77030, USA; auosef@houstonmethodist.org (A.U.); rmghobrial@houstonmethodist.org (R.M.G.); 2Department of Surgery, Houston Methodist Hospital, Houston, TX 77030, USA; 3MD Anderson Cancer Center, Department of Genetics, University of Texas, Houston, TX 77030, USA; 4Department of Regenerative Medicine and Cell Biology, Military Institute of Hygiene and Epidemiology (WIHE), 01-163 Warsaw, Poland; jacek.kubiak@univ-rennes1.fr; 5Cell Cycle Group, Faculty of Medicine, Institute of Genetics and Development of Rennes (IGDR), University Rennes, CNRS, UMR 6290, 35043 Rennes, France

**Keywords:** placenta, syncytin, spike protein, retrovirus, SARS-CoV-2, COVID-19 vaccine

## Abstract

**Simple Summary:**

The anti-vaccination movement claims an alleged danger of the COVID-19 vaccine based on the presupposed similarity between syncytin, which plays a role in human placentation and the SARS-CoV-2 spike protein. We argue that because of very low sequence similarity between human syncytin-1 and the SARS-CoV-2 S protein, it is unlikely that any S protein-specific SARS-CoV-2 vaccine would generate an immune response which would affect fertility and pregnancy. However, further evaluation of potential impacts of COVID-19 vaccines on fertility, placentation, pregnancy and general health of mother and newborn is required.

**Abstract:**

Human placenta formation relies on the interaction between fused trophoblast cells of the embryo with uterine endometrium. The fusion between trophoblast cells, first into cytotrophoblast and then into syncytiotrophoblast, is facilitated by the fusogenic protein syncytin. Syncytin derives from an envelope glycoprotein (ENV) of retroviral origin. In exogenous retroviruses, the envelope glycoproteins coded by *env* genes allow fusion of the viral envelope with the host cell membrane and entry of the virus into a host cell. During mammalian evolution, the *env* genes have been repeatedly, and independently, captured by various mammalian species to facilitate the formation of the placenta. Such a shift in the function of a gene, or a trait, for a different purpose during evolution is called an exaptation (co-option). We discuss the structure and origin of the placenta, the fusogenic and non-fusogenic functions of syncytin, and the mechanism of cell fusion. We also comment on an alleged danger of the COVID-19 vaccine based on the presupposed similarity between syncytin and the SARS-CoV-2 spike protein.

## 1. Introduction

The mammalian placenta evolved around 150–200 million years ago. The placenta facilitates the exchange of nutrients, gases, and metabolic end products between mother and fetus. It also produces hormones regulating fetal and maternal physiology and provides immunotolerance toward the paternal (thus, alien to the mother) component of the embryo. In some mammals, including humans, the syncytialization process and the formation of invasive syncytiotrophoblast allows an intimate contact with the uterine endometrium and the formation of the placenta. It has to be noted here that this invasive capacity is rare in most mammals, and besides the transfer of compounds between maternal and fetal blood (humans and some other mammals), other functions of the syncytiotrophoblast are metabolite transfer and hormone production. Although the common understanding is that the placenta is a structure unique to the placental mammals, the placenta equivalent evolved independently multiple times in many groups of nonmammalian vertebrates [1,2]. For example, some ground sharks have a yolk sac placenta that forms through close apposition of an emptied (after the embryo had consumed all nutrients) yolk sac with the vascularized oviduct [1,3]. In some stingrays, the wall of the uterus forms multiple villi, which contact and feed the embryo [1,4]. In potbellied sea horses, eggs deposited in the father’s pouch stick to and are nourished by vascularized folds of pouch epithelium [1,5]. In marsupial frogs, eggs are deposited in the pouch on the mother’s back, and developing embryos are fed through a “gill placenta” formed between the pouch epithelium and specialized gills of the embryo [1,6,7,8]. In many reptiles, the embryonic chorioallantois (a vascularized fetal membrane composed of the fused chorion and wall of the allantois) forms a close connection with the uterine epithelium or endothelium of the blood vessels [1,9,10]. These placenta equivalents are extremely efficient in feeding and supporting the development of the embryos; in some of these species, the weight of the newborn is 500–3000 times higher than that of the fertilized egg [1,4,11].

## 2. Types of Mammalian Placenta

The mammalian placenta develops shortly after implantation of the blastocyst into the endometrium (inner epithelial layer with its mucous membrane) of the uterus. First, the pluripotent cells of the blastocyst segregate into the outer layer (trophoblast, trophectoderm) and the inner cell mass, which differentiate into the hypoblast and epiblast. The hypoblast participates in the formation of the yolk sac that gives rise to the chorion. The epiblast differentiates into the ectoderm, mesoderm, and endoderm of the embryo proper. The trophoblast forms the outer layer of the placenta, which is underlined by the cytotrophoblast layer and is covered by the syncytiotrophoblast that forms from the fusion of the cytotrophoblast-derived cells. Depending on the species, the trophoblast either closely apposes or penetrates (invades) to a different degree the uterine wall [12,13]. In humans, at embryonic day six (E6), the mononuclear cells of the cytotrophoblast fuse to form the invasive syncytium (syncytiotrophoblast) that allows the attachment of the embryo to the uterine endometrium. At later stages of placentation the mononuclear “extravillous” trophoblasts comprise the primary invading trophoblasts. However, an invading syncytialized trophoblast is not a conserved feature of placentation in all mammals.

With the exception of the egg-laying echidna and platypus (monotremes), all mammals, including marsupials, which are often classified as nonplacental mammals, have placenta [1]. However, in contrast to other mammals, the marsupial placenta is short-lived and the newborns, which migrate to the pouch, are very underdeveloped [1]. Additionally, the early specification of cell lineages in the marsupial blastocyst differs from other mammals; there is no inner mass, and the trophoblast (that forms at the pole of the blastocyst opposite to the rest of the embryo) together with the yolk sac endoderm forms the yolk sac placenta with a different, between species, degree of invasiveness of trophoblast into the uterine wall [1,14].

There are many different strategies to classify placentas, one of which is the degree of separation between maternal blood and trophoblast. Based on this, the eutherian mammals (commonly classified as placental mammals) have four main types (with a variety of subtypes) of the placenta [1,15,16,17]: 1. the epitheliochorial (noninvasive) placenta (hippopotamuses, pigs, horses), in which the trophoblast closely apposes but does not penetrate the uterine wall; 2. the endotheliochorial placenta (carnivores), where the trophoblast invades the uterine wall and reaches but does not penetrate blood vessels; 3. the hemochorial placenta (humans, apes, monkeys, and rodents) in which the trophoblast penetrates the capillaries and is in direct contact with maternal blood, and 4. the synepitheliochorial (cotyledonary) placenta (ruminants) that instead of a single large area has several smaller sites (placentomes). In this type of placenta the trophoblast, besides the mononuclear and binuclear cells, also contains cells with three nuclei. These trinuclear cells are believed to derive from the fusion of the uterine cells with binuclear trophoblast cells [1,16]. If this is the case, then this type of placenta is exceptional because its formation involves the heterologous fusion between embryonal and maternal cells. Studies of hyenas, which are the only carnivore with an invasive hemochorial placenta, showed, besides the common to all carnivores syncytin-Car1 gene, the expression of the hyena-specific syncytin gene Hyena-Env2. This gene is expressed at the maternal-fetal interface of placenta but is non-fusiogenic. The same studies also identified the syncytin-Mab1 gene in the viviparous Mabuya lizards, which have a human-like placenta. This gene is fusogenic and is expressed in a fused cell layer of the lizard placenta [18,19]. Although it has been hypothesized that the noninvasive placenta, as potentially the least effective in nourishing the fetus because of the lack of direct contact with maternal blood, is evolutionarily most ancient, and thus primitive, this is not likely the case. The comparative analysis of placenta within the mammalian phylogenetic tree indicates that the ancestral (primitive) placenta was the invasive-type and that the noninvasive placenta is a derived form that evolved several times independently in different mammalian groups [1,20]. Additionally, Wildman et al. (2006) [20] suggest that the ineffectiveness of the noninvasive placenta is very much exaggerated. They argue that the small (a few micrometers) distance between the fetal and maternal tissue does not really hinder the exchange of small molecules, and the exchange of the macromolecules is supported by histotrophic nutrition—the endocytosis of uterine gland secretion components by specialized cells of the trophoblast [21]. Additionally, the noninvasive placenta has several advantages: the lack of contact with the maternal blood limits a potential transmission of maternal pathogens to the fetus; limits the exchange of maternal and fetal cells, lowering the immune response against the paternal component of the fetus; and drastically reduces injury to the uterine wall during birth detachment of the placenta [1].

## 3. Exaptation of Retroviral Genes for Placental Function

The paleovirological analyses of fossils sequences and available genomic sequences from different organisms indicate that the genome of eukaryotic cells was colonized many times during evolution by foreign genetic entities called the mobile or transposable elements (TEs) and newly described, and less frequent, the endogenous viral elements (EVEs) [22,23,24]. The EVEs either mutate, become silent or transform into useful genes used by the host organisms for different purposes—the process known as exaptation (a shift in the function of a trait during evolution). Around 45% of the mammalian (human and mouse) genome is composed of TEs and EVEs. There are two main categories of TEs; category 1 includes the long terminal repeat (LTR) and non-LTR retrotransposons, and endogenous retroviruses (ERVs), while category 2 includes DNA transposons. The endogenous retroviruses permanently integrated into the host genome are derived from the exogenous retroviruses which infected the host cells. The exogenous retroviruses are composed of the lipid/glycoprotein envelope, various proteins, and RNA. The main proteins encoded by the retroviral RNA are the group-specific antigen (gag) proteins, which play a role in packaging RNA and virus assembly; pol proteins participating in the synthesis and integration of viral DNA into the host genome; proteases (pro), which process the gag and pol proteins, and the envelope (*env*) proteins facilitating virus entry into the host [25]. Retroviral RNA also encodes the reverse transcriptase and integrase enzymes. Reverse transcriptase copies the retroviral RNA into cDNA that is subsequently incorporated into the genome of a host cell by integrase [26,27,28]. At this point, the integrated viral DNA is referred to as a provirus. The host cell transcribes and translates viral genes producing all proteins needed for the new copies of the mature and infectious virus. Those exogenous retroviruses that had integrated into the genome of the germline cells are inherited by the next generation and become, as the endogenous retroviruses (ERVs), a permanent fixture of the host genome. Between 5% and 8% of the human genome is made of such endogenous retroviral DNA [22]. Although during evolution, the majority of TEs underwent rearrangements and lost their original functions/coding capacity, the evolutionarily youngest TEs, or those which were domesticated for the fulfillment of the host’s critical functions, are still functional. An example of such necessary for the host retroviral proteins are retroviral *env* genes. In the exogenous retroviruses, the envelope glycoproteins (Env) coded by the *env* genes allow the fusion of the viral envelope with the host cell membrane and entry of the virus into the host cell. During mammalian evolution, the *env* genes have been repeatedly, and independently, captured by various mammalian species to facilitate the formation of the placenta [22].

## 4. Endogenous Retroviral Gene Function in the Placenta

Screening of the human genome for the presence of the retroviral *env* genes identified 18 genes with the long open reading frame [22,29,30].One of these genes, the human endogenous retrovirus W (HERV-W) belongs to the tryptophan tRNA (W) gene family and another, the HERV-FRD, belongs to the human endogenous retrovirus dihydrofolate reductase (FRD) gene family [22,31,32,33,34]. Both these genes are specifically expressed in the placenta and the proteins they encode cause the cell fusion and the formation of syncytium in cultured cells, and the syncytiotrophoblast in the placenta [22]. Because of the syncytium-promoting function, they were given the name syncytin (HERV-W, the syncytin-1 and HERV-FRD, the syncytin-2). The syncytin-1 binds the Na-dependent neutral amino acid transporter 2 ASCT2 (SLC1A5), (Figure 1 and Figure 2), while syncytin-2 binds the major facilitator superfamily domain-containing protein 2 MFSD2 (also called a sodium-dependent lysophosphatidylcholine symporter 1 [31,35,36].

The left panel shows the main domains (marked in different colors) of the syncytin-1 molecule. The right panel shows six consecutive steps of syncytin-1 molecule transformation during membrane fusion. The membrane layers depicted in the drawings belong to two different fusing cells. The colors of the syncytin-1 molecule correspond to those shown in the left panel. The binding of the receptor-binding domains (RBs) to the receptor (ASCT-2) breaks the disulfide bonds and unfolds the fusion peptide that becomes inserted into the membrane. This is followed by the positional changes of heptad repeats -1 and -2 domains. The final steps consist of membrane apposition and bending (adapted from [36]).

The interaction between the syncytin-1 and its receptor ASCT-2 causes structural reorganization of the syncytin molecule, such as unfolding the fusion peptide that penetrates the cell membrane of one of the fusing partners. It also changes the reorganization of actin filaments, which give fluidity/stiffness to the fusing membranes. The hemifusion (the scission of the membrane in one partner) is followed by the formation of the fusion pore, which connects the cytoplasm of both fusing partners. The circles in the drawing represent two fusing cells.

The reduced levels of syncytin expression in trophoblast cells causes various placental pathologies. Paleovirological analyses established that the syncytin-1 is around 30 million years old and the syncytin-2 around 45 million years old. The conservation of the open reading frame in the syncytin genes of various mammalian species, a low mutation rate, and a low level of polymorphism among humans, indicates that these genes played an essential role in mammalian evolution [22,37].

## 5. Other Syncytins

Besides human syncytin-1 and syncytin-2, there are other mammalian syncytins. The mouse genome contains syncytin-A and syncytin-B, which are different than human syncytin-1 and -2, but have the same function and retroviral origin, and were integrated into the mouse genome around 25 million years ago [22,38,39,40]. Syncytin-A and syncytin-B induce cell–cell fusion in the in vitro assay [38]. Knock-out studies showed that syncytin-A is crucial for mouse placentation and is required for the formation of the murine syncytiotrophoblast [41,42]. The orthologous of syncytin-A and -B are also present in other rodents, including rat, mole rat, gerbil, vole, and hamster [22,40]. The rabbits and hares have another syncytin gene, syncytin-Ory1, which was acquired around 12 million years ago, while the carnivores have syncytin-Car1 that, because it was acquired around 80 million years ago, is the evolutionarily oldest mammalian syncytin gene, and the higher ruminants have the syncytin-Rum1 gene [22,43,44]. Studies of syncytin-Ory1 showed that its receptor is identical to the syncytin-1 receptor ASCT2. The expression of Ory-1, in the junctional zone between the placental lobe and the maternal decidua, suggests its role in the formation of the syncytiotrophoblast [45]. Syncytin-Mar1 is another syncytin expressed in the placenta of the woodchuck Marmota monax. Syncytin-Mar1 is evolutionarily unrelated to the syncytin genes present in primates, muroids, carnivores, and ruminants but has the same fusogenic function [46].

## 6. Hypothetical Role of Syncytin-1 in Fertilization

The syncytin-1 is also present in the human gametes and, although there are no functional studies, hypothetically it may be involved in the gamete fusion during fertilization (Figure 3).

During fertilization, there is a fusion between the sperm and the egg membrane (oolemma). The syncytin-1 and its receptor (ASCT-2) are present in the sperm and oocytes/eggs. Sperm is surrounded by the plasma membrane. The sperm head contains the nucleus and the acrosome, which is surrounded by the outer and inner membrane and contains proteolytic enzymes. The midpiece region of the sperm contains centriole, which forms the axoneme (microtubules), which in turn is surrounded by a sheath of mitochondria. After recognition of egg presence, the sperm undergoes an acrosome reaction that fuses the outer acrosomal membrane with the sperm plasma membrane, releasing the proteolytic enzymes. In the next fusion event, the acrosomal inner membrane fuses with the oolemma, permitting the transfer of sperm genetic material to the egg cytoplasm. Although there are no functional studies supporting this, hypothetically, both of these fusion events may require the fusogenic role of syncytin [47,48,49].

The tip of the sperm head contains a cap-like structure called the acrosome that contains proteolytic enzymes and it is surrounded by the outer (at the top) and the inner (at the bottom) membrane. When the spermatozoon reaches the egg vicinity, the zona pellucida of the egg induces in the sperm the acrosome reaction (AR), which is the fusion of the outer acrosome membrane with the membrane of the sperm’s head, which concentrates the zona pellucida binding proteins at the apical surface of its head and releases proteolytic enzymes. This is followed by the adhesion and penetration of zona pellucida, the adhesion and fusion of the inner acrosomal membrane with the egg membrane (oolemma). This fusion deposits the genetic material of the mother and father in a single zygote. Thus, during fertilization, there are two independent membrane fusion events: the fusion (during AR) of the outer acrosomal membrane with the sperm plasma membrane, and the fusion of the inner acrosomal membrane with the egg membrane, when the syncytin, as a fusogenic protein, is most probably involved. Quantitative RT-PCR studies of the syncytin-1 expression in the human gametes showed that syncytin-1 is present in the sperm head, while its receptor ASCT-2 is expressed in the acrosome and the sperm tail. The ASCT-2 is also expressed in the oocytes/eggs [47,48]. These findings suggest that the lack or reduced expression of syncytin-1 and its receptor may lead to fertilization failure and open new avenues for the treatment of infertility. However, there is no evidence that syncytin-related genes are involved in fertilization in any other species. In mice lacking syncytin A and B, the egg/sperm fusion occurs normally as evidenced by the resulting pregnancy, which indicates that, at least in mice, these syncytins are not required for fertilization [41,42]. Syncytin is not the only fusogenic protein that could be involved in fertilization and act next to fusion-required proteins IZUMO1 at the sperm membrane, and JUNO at the oocyte membrane [49]. The possible role of syncytin in mammalian fertilization resembles the role of the hapless 2/generative cell specific1 or HAP2/GCS1 gene of viral origin. This gene is involved in the fertilization in plants (*Arabidopsis thaliana* or trumpet lily *Lilium longiflorum)*, unicellular algae (*Chlamyodomonas reinhardii* and *Gonium pectoral)*, protists (*Tetrahymena thermophila*, *Trypanosoma cruzi* and *Plasmodium falciparum*), hydras, and the honeybee [50,51]. However, this gene is absent, or not found in vertebrate species including mammals. The proteins coded by syncytin and HAP2/GCS1 share the tridimensional structure with the viral fusogenic proteins. HAP2/GCS1 ancestor genes were captured by the genome of predecessors of the organisms listed above, most probably more than 80 million years ago becoming the EVE. The viral origin of syncytins and HAP2/GCS1suggests the multiple and independent incorporation events of viral genes by different genomes during a whole course of evolution. The intriguing hypothesis is that the early exaptations of these viral genes for reproductive purposes may even be at the origin of the sexual reproduction of eukaryotes.

## 7. Syncytin Role in Cancer Cell Fusion

Many studies showed that various cancer cells express syncytin and/or syncytin receptors and can fuse with normal healthy cells. The results of such fusions differ between cell types and are either beneficial (suppression of tumorigenicity) or detrimental (increased proliferation activity, genetic instability, and malignant transformation). For example, the level of syncytin expression in breast cancer patients is a positive prognostic indicator for recurrence-free survival. It is believed that the syncytin-mediated fusion between the healthy endothelial cells and breast cancer cells suppresses the proliferation potential of the latter. In contrast, the fusion between spleen cells and myeloma cells results in the formation of hybrid cells with an unlimited proliferative ability, and the fusion between melanoma cells and macrophages enhances malignant phenotype [52,53].

## 8. How Syncytin Facilitates Cell Membrane Fusion?

Fusion is a local rupture of continuity of the membrane lipid bilayers in the fusing cells and their subsequent rejoining [54]. By facilitating fusion, syncytin belongs to the group of proteins called the fusogens (Figure 1 and Figure 2). The fusion can occur either by the unilateral mechanism when the cell membrane of one of the partners contains fusogen, the bilateral homotypic mechanism when both cells contain the same fusogen, or the bilateral heterotypic mechanism, when the fusogen is different in each partner. In general, the fusogen brings the lipid bilayers of two cells into immediate contact, and catalyzes the formation of energy-intensive fusion intermediates, and the formation of a fusion pore [55,56]. Although the fusion events differ between the cell types and circumstances (embryo development, disease, tissue injury, regeneration, repair) they all have common features: the cells have to be fusion-competent, their membranes must adhere, then membranes fuse, and finally there is the post-fusion resetting of the membrane structure. Based on the studies of the myotube formation, Zhou and Platt [57] proposed the model of cell fusion in which the lipid rafts of the opposing membranes recruit and align the adhesion molecules, while the cortical actin serves as a supportive platform. In the next step, the interaction between the adhesion proteins causes rearrangement of the actin cytoskeleton. This results in the dispersion of lipid rafts and a direct contact between the opposing phospholipid bilayers. The force generated by actin polymerization leads to the formation of the fusion pores (channels between the fusing cells), which subsequently expand [58]. The molecular mechanism of syncytin-dependent fusion is probably identical or similar to that mediated by the class I viral fusogens (such as HIV ENV). These viruses’ fusogen proteins are organized into trimers, each containing two α helices and an amphiphilic fusion peptide. The conformational change of the trimer results in the formation of a stiff-coiled coil structure with exposed fusion peptide at the N terminus. The fusion peptide is inserted into the membrane of the partner cell, leading to the tightening of the membrane and the formation of the fusion pore [55].

## 9. Syncytin Functions beyond Cell Fusion—The Immunomodulatory Functions

Syncytin-1 shares high sequence similarity to the multiple sclerosis retrovirus-like particle (MSRV) envelope protein that is involved in the development of multiple sclerosis. The abnormal expression of syncytin-1 is also one of the triggering factors in bipolar disorder and schizophrenia [59,60,61,62,63]. In neuropsychological diseases, syncytin-1 mediates the chronic inflammation in the nervous system, which can cause neuronal injury and/or damage to the brain microvasculature, cerebral blood flow, and the blood–brain barrier. Several studies indicated that syncytin-1 induces proinflammatory cytokines via toll-like receptor 4 (TLR4), NF-κB, and the glycosylphosphatidylinositol-anchored membrane protein CD14 (that functions as a pattern recognition receptor with the extracellular domain of TLR4), signal the transduction pathway [62,63].

It has been shown that the expression of syncytin confers resistance to the spleen necrosis virus infection [64]. This suggests that syncytin may prevent infection with other retroviruses and also inhibit retroviruses’ transmission through the placenta [56]. Syncytin is also involved in immunosuppression through its immunosuppressive domain (ISD) [65]. Recent studies showed that syncytin-2 suppresses the function of T cells. Treatment of activated Jurkat T cells and peripheral blood mononuclear cells (PBMCs) with monomeric or dimeric Syn-2 ISD peptide changed the phosphorylation of ERK1/2 MAP kinases, and reduced Th1 cytokine production and T cell activation [66].

## 10. The Limited Similarity of Syncytin to SARS-CoV-2 Spike Protein

Currently, there are more than 170 COVID-19 vaccines undergoing clinical trials. There are four main types of COVID-19 vaccines: whole virus, nucleic acid (RNA or DNA), viral vectors, and protein subunits. Whole virus vaccines use either live but weakened virus or virus the nucleic acid of which has been inactivated to prevent replication. The live weakened virus-based vaccine can sicken people with a weak immune system. Nucleic acid vaccines use viral RNA or DNA that upon delivery into the cells produce the antigen. Viral vectors vaccines usually use adenovirus to introduce viral genetic information to produce antigen. Protein subunit vaccines introduce a fragment of the viral protein (antigen) [67,68]. The entry of the SARS-CoV-2 virus (causing the COVID-19 pandemic) is facilitated by the spike protein on the surface of the virus [69,70,71,72]. Thus, the viral spike protein is the primary target antigen for many manufacturers of COVID-19 vaccine. The currently licensed Moderna and Pfizer–BioNTech vaccines use mRNA coding for the spike protein enclosed within the nanolipid particles, while the Oxford–AstraZeneca vaccine uses a replication-deficient adenovirus with inserted SARS-CoV-2 spike protein sequences. Although the unprecedentedly fast development of the COVID-19 vaccines allows for the immediate vaccination of millions of people worldwide and promises the end to this devastating pandemic, there are some skeptics and anti-vaccination movements which herald the danger of the vaccine. They claim that the COVID-19 vaccine may cause female and male infertility, problems in pregnancy, cancer, etc. These claims are based on the supposed similarity between the SARS-CoV-2 spike protein and the syncytin protein that, as described above, probably participates in the fusion of gametes during fertilization, the formation of the placenta during pregnancy, and the fusion of cells in certain cancers. Thus, according to the vaccination skeptics, the immune response directed against the spike protein will also target and disrupt the syncytin and its related functions. The syncytin gene (the retroviral origin envelope ENV gene) is located on human chromosome number 7 (7q.21.20). The syncytin protein has 538 amino acids. Recently studies showed that the regions of N- and C-terminal heptad repeats NHR (41 aa) and CHR (34 aa) in the S2 domain of the SARS coronavirus (SARS-CoV) spike protein and syncytin share some very limited similarities [34]. However, the comparison of human syncytin-1 (538aa) and SARS-CoV-2 spike protein (1273aa) indicates that they share only a few amino acids, and that a comparably low degree of similarity can be found between the viral spike protein and any other proteins in the human body. The comparison of these two sequences at the amino acid level by the Blast program shows zero homology, and when the search is limited to the small stretches of similarity there are only two 2-amino acid stretches identical between these two proteins (Figure 4), [73,74].

The Blast comparison of these two sequences does not show any similarities. However, the search for the similarity between 5-amino acid stretches (marked in yellow) shows two 2-amino acid identities (marked in red). Such very limited similarity is very unlikely to cause cross-reactivity between anti-SARS-CoV-2 antibodies and the human syncytin-1 protein.

The empirical bioinformatic studies showed that the protein pairs which share 8 amino acid identities, but not > 35% identity over 80 amino acid stretches, are not cross-reactive [75]. Although there are no published clinical data on the safety of the COVID-19 vaccine for pregnancy, placentation, and fertility, there are very strong indications that pregnant women should be the first candidates for preventative measures, such as COVID-19 vaccination [76,77]. Pfizer/BioNTech are conducting animal studies on the effects of the COVID-19 vaccine on pregnancy, and unpublished reports indicate that the vaccine is safe. Moderna performed similar studies using a rat model and concluded that there were no adverse effects on female reproduction, fetal development, or postnatal development. During the Pfizer/BioNTech vaccine clinical trials, 23 women became pregnant and so far no adverse effects have been reported [78]. Both the Centers for Disease Control and Prevention (CDC) and The American College of Obstetricians and Gynecologists (ACOG) recommend that the COVID-19 vaccine should not be withdrawn from pregnant and lactating women [79,80]. Further evaluation of potential impacts of COVID-19 vaccines on fertility, placentation, pregnancy and general health of mother and newborn is required. However, on the basis of the very low sequence similarity between human syncytin-1 and the SARS-CoV-2 S protein, it is considered to be unlikely that any S protein-specific SARS-CoV-2 vaccine would generate an immune response which is cross-reactive with syncytin 1 and this way affect fertility and pregnancy.

## 11. Conclusions

Human syncytin-1 plays a role in human placentation. Syncytin-1 has a retroviral origin and is slightly similar to the spike protein expressed on the surface of SARS-CoV-2. The similarity between sycytin-1 and SARS-CoV-2 spike protein is very limited. It is very unlikely that any spike protein-specific SARS-CoV-2 vaccine would generate an immune response which is cross-reactive with syncytin 1 and affect fertility and pregnancy.

## Figures and Tables

**Figure 1 biology-10-00238-f001:**
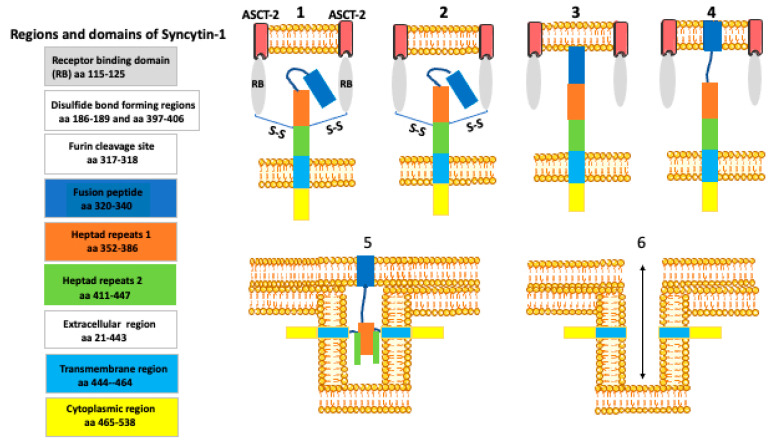
The main regions and domains of syncytin-1 and its conformational transformation during the membrane fusion.

**Figure 2 biology-10-00238-f002:**
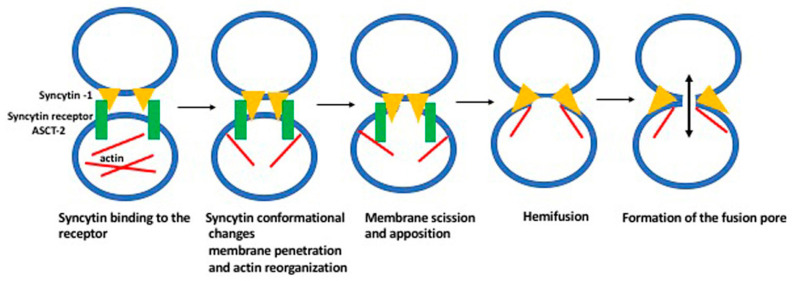
The consecutive steps in cell fusion.

**Figure 3 biology-10-00238-f003:**
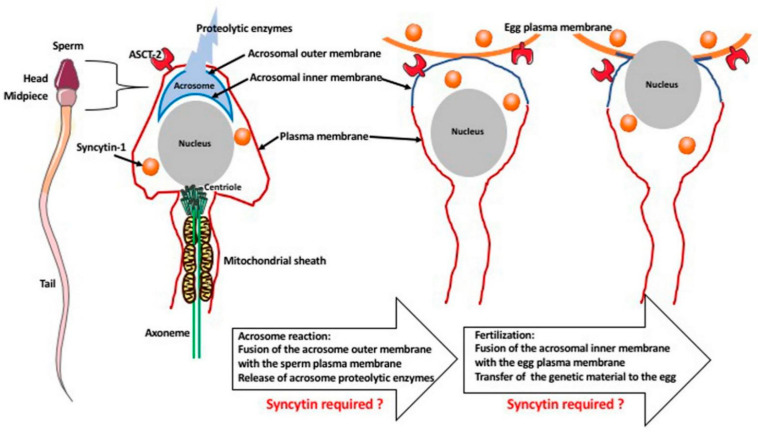
Membrane fusion during fertilization.

**Figure 4 biology-10-00238-f004:**
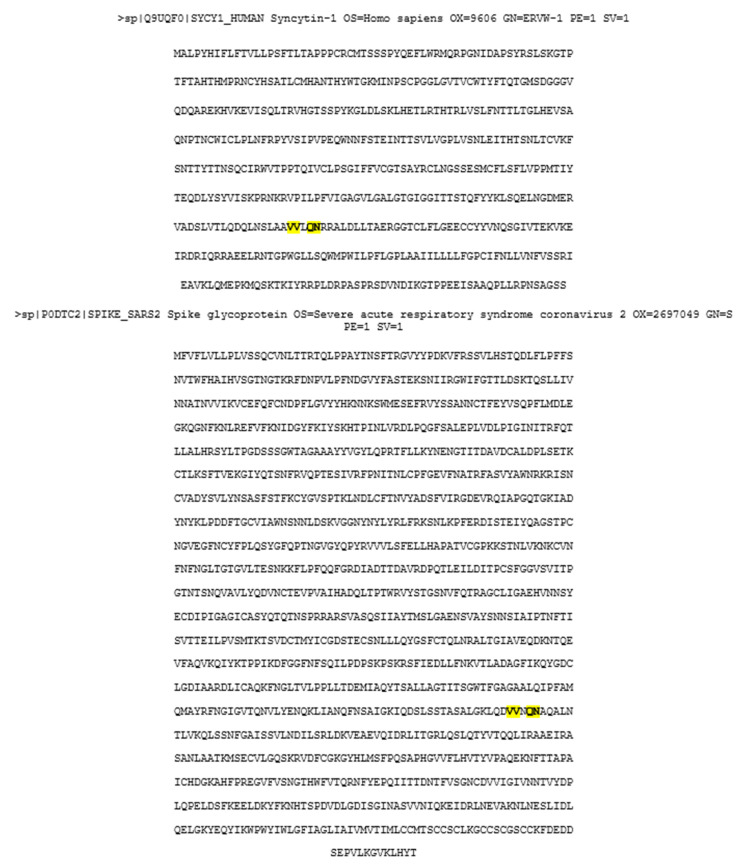
The comparison between amino acid sequences of human syncytin-1 and the spike protein of SARS-CoV-2.

## Data Availability

Not applicable.

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
