# Peer review of "Exaptation of Retroviral Syncytin for Development of Syncytialized Placenta, Its Limited Homology to the SARS-CoV-2 Spike Protein and Arguments against Disturbing Narrative in the Context of COVID-19 Vaccination"

_biology, 2021, doi:10.3390/biology10030238_

Round 1

Reviewer 1 Report

In the present manuscript, the authors have reviewed the exaptation of retroviral genes for mammalian placenta. They compared the peptide sequence of the spike protein from SARS-CoV-2 to that of the human Syncytin-1. This comparison indicated nearly no sequence similarities between the two cases, and thus the authors concluded that COVID-19 vaccines will not affect pregnancy, et.al. The question raised by the authors itself is interesting and has the urgency under the current global pandemic. However, the review needs major revision before publication.

1) One of the main goals of the paper is to clarify whether COVID-19 vaccines can recognize syncytin proteins, and then to conclude the effects of these vaccines on pregnancy. The work should be of broad interest, and thus the mechanisms behind COVID-19 is important. Currently, there are different types of COVID-19 vaccines (like the m-RNA vaccines and the inactivate ones), the authors should describe them in the paper.

2) All figures provided in the manuscript are rather poorly made, it is hard to read the texts on them. The authors should increase the quality of figures.

Author Response

Review 1

English language and style

( ) Extensive editing of English language and style required
( ) Moderate English changes required
(x) English language and style are fine/minor spell check required
( ) I don't feel qualified to judge about the English language and style

Comments and Suggestions for Authors

In the present manuscript, the authors have reviewed the exaptation of retroviral genes for mammalian placenta. They compared the peptide sequence of the spike protein from SARS-CoV-2 to that of the human Syncytin-1. This comparison indicated nearly no sequence similarities between the two cases, and thus the authors concluded that COVID-19 vaccines will not affect pregnancy, et.al. The question raised by the authors itself is interesting and has the urgency under the current global pandemic. However, the review needs major revision before publication.

  • One of the main goals of the paper is to clarify whether COVID-19 vaccines can recognize syncytin proteins, and then to conclude the effects of these vaccines on pregnancy. The work should be of broad interest, and thus the mechanisms behind COVID-19 is important. Currently, there are different types of COVID-19 vaccines (like the m-RNA vaccines and the inactivate ones), the authors should describe them in the paper.

Response: As requested, we added the information on the types of vaccines, including the currently used COVID-19 vaccines, and added the references

2) All figures provided in the manuscript are rather poorly made, it is hard to read the texts on them. The authors should increase the quality of figures.

Response: we increased the quality of figures and also changed the font size of the figure texts

Reviewer 2 Report

This review article entitled “exaptation of retroviral syncytin for development of placental mammals, its limited homology to SARS-CoV-2 spike protein, and false narrative during COVID-19 pandemic” focuses on the origin of syncytins as endogenous retroviral elements and regulators of different types of placentation. The review article also briefly compares the amino acid sequences of syncytin-1 with SARS-CoV-2 spike protein. A significant limitation in this review is that the authors try to capture the general importance of syncytialization for placentation in all mammals, but many mammals do not possess a syncytialized trophoblast (especially an invasive one) or rely on cell fusion for placental development, so it is not accurate to suggest that syncytin expression and syncytialization is a requirement for placentation in all mammals. The review is also disjointed in parts. For instance, different types of placentas or “placenta-like structures” are discussed in marsupials, fish, and different mammals, which while interesting, has nothing to do with syncytin-1, exaptation of syncytins, or homology of syncytins with SARS-CoV-2. The review also ignores all other syncytins except syncytin-1, which is only involved in human and closely-related primate placentation and not involved in placentation of most mammals. This is not consistent with the objective of the review to discuss “exaptation of retroviral syncytin for development of placental mammals”. Other syncytins are only mentioned when convenient (for instance, in the last sentence of the immunomodulatory paragraph where syncytin-2 is discussed). More specific comments for improvement are provided:    

Major comments:

  • The introduction and first few sections have generalizations which are not accurate, such as:
    • “In the simplest term, the mammalian placenta is a temporary fetal/maternal organ formed through the tight apposition of the specialized fetal syncytial tissue with the wall of the uterus.” This is an incorrect generalization. As an example, in human placenta, extravillous trophoblasts ultimately form the major interface with the decidua (endometrial stroma, not wall of the uterus). Other species have drastically different types of placentation, only some of which appose the wall of the uterus and few of these involve syncytin-generated syncytia.
    • “The syncytialization process is essential for placentation because only the syncytial trophoblast has the invasive capacity allowing intimate contact with the uterine endometrium and the formation of the placenta. Thus, the cell fusion occurring during syncytialization plays a pivotal role in placenta formation, and consequently, in the development and reproduction of placental mammals.” This is true in human placentation during the first few days of implantation, but at later stages mononuclear “extravillous” trophoblasts comprise the primary invading trophoblasts. An invading syncytialized trophoblast is also not a conserved feature of placentation in mammals.
    • About half the introduction is a description of nonmammalian placentas. What is the point of including these in the introduction? Do these placenta-like structures undergo cell fusion? Do they express syncytins?
  • There is a large section on the role of syncytin-1 in fertilization. To date, there has only been one report that synctyin-1 and its receptor is expressed in human gametes, and no functional studies (the current review includes two articles, but one is a review article). There is also no evidence that syncytin-related genes are involved in fertilization in any other species. For example, Syncytin-A and Syncytin-B deficient mouse embryos implant and are viable up until at least mid-gestation. Thus, the notion that syncytins are absolutely required for different phases of egg-sperm fusion (as suggested in Figure 3), and that they may have instigated fertilization in all eukaryotes, should be removed altogether or relegated to a sentence or two speculating that syncytins could be involved in fertilization in some species but functional studies are required.
  • It is surprising that syncytiotrophoblast is emphasized as an invasive structure, which is rare in most mammals, and it is not mentioned for its primary function as lining the exchange surface regulating transfer of substances between maternal and fetal blood in some placentae (humans and a few other mammals), or for its critical role in hormone production and metabolism.
  • The section on the syncytin-1 comparison to COV-2 reads more like an opinion piece than a scientific report, with language that includes “misinformed skeptics”, and “erroneous claims”. Given that more than half the title is devoted to homology between syncytin-1 and Spike protein, it was surprising that the article only contains a BLAST search comparing amino acid sequences and a single paragraph rebuking skeptics. For instance, there is no discussion on the similar genetic elements in which many of the concerns of similarity are based, three-dimensional structure of each protein and their different affinities for unique receptors, or how there are differences between CoV and CoV2 spike proteins, the former having previously identified similarities with Syncytin-1.
  • In line with the above statement, the conclusions of this review article are based on opinion: “Thus, in summary, the very limited similarity between the SARS-CoV-2 spike protein and human syncytin-1 assures that the COVID-19 vaccines are very specific, and they will not recognize (or destroy) syncytin protein, and will not affect, fertility, pregnancy, or cause cancer.” While this is certainly hopeful, there is no way for the authors to be able to make these claims based only on a BLAST search comparing the CoV-2 spike protein and syncytin-1. It would be more accurate to assert that based on the limited sequence similarity between these proteins, it is unlikely that any vaccine targeted against the CoV-2 spike protein would generate an immune response against syncytin-1. The review could also benefit from including some sentences about the current evidence suggesting that those who have had COVID19 or have received vaccines are fertile, able to conceive, and carry a pregnancy, which indicates that the vaccine (or virus) is not likely to impact these process in the short-term. However, it is prudent to continue to evaluate any potential impact of COVID19 and/or vaccines on fertility and health.

Minor comments:

  • “Have four main types (with a variety of subtypes) of the placenta”…there are different ways to classify placentas, of which the degree of layers separating maternal blood and trophoblast is one strategy.
  • “Although it would seem that the noninvasive placenta, as potentially the least effective in nourishing the fetus because of the lack of direct contact with maternal blood, is evolutionary most ancient, and thus primitive, this is not the case.” This is a hypothesis based on some excellent comparative placentation studies, so the wording should be changed to reflect that “this does not appear to be the case” or “this is not likely the case”.  
  • “Thus, the COVID-19 vaccines are targeting the viral spike protein.” – is there a reference for this statement? Maybe soften this statement to enable some flexibility for different vaccine strategies, such as “the viral spike protein is the primary target antigen for many vaccine manufacturers”
  • The review is not comprehensive, consisting of just over 60 references, and requires significant editing. “The” is overused when describing terms, for example (page 7): “The fusion is the local rupture…” and “…when there the fusion is different…”. There are also many awkward sentences. As one example, in Section 3: “Although during the evolution, the majority of TEs rearranged and lost their original functions/coding capacity, the evolutionarily youngest or/and domesticated for the fulfillment of the critical for the host organism functions are still functional. An example of such necessary for the host retroviral proteins are retroviral env genes”.

Author Response

Review 2

English language and style

( ) Extensive editing of English language and style required
(x) Moderate English changes required
( ) English language and style are fine/minor spell check required
( ) I don't feel qualified to judge about the English language and style

Comments and Suggestions for Authors

This review article entitled “exaptation of retroviral syncytin for development of placental mammals, its limited homology to SARS-CoV-2 spike protein, and false narrative during COVID-19 pandemic” focuses on the origin of syncytins as endogenous retroviral elements and regulators of different types of placentation. The review article also briefly compares the amino acid sequences of syncytin-1 with SARS-CoV-2 spike protein.

A significant limitation in this review is that the authors try to capture the general importance of syncytialization for placentation in all mammals, but many mammals do not possess a syncytialized trophoblast (especially an invasive one) or rely on cell fusion for placental development, so it is not accurate to suggest that syncytin expression and syncytialization is a requirement for placentation in all mammals.

Response: We agree with the reviewer that the syncytialization does not occur in all mammals, and we changed the title, abstract and the introduction to more accurately describe the article premise.

The review is also disjointed in parts. For instance, different types of placentas or “placenta-like structures” are discussed in marsupials, fish, and different mammals, which while interesting, has nothing to do with syncytin-1, exaptation of syncytins, or homology of syncytins with SARS-CoV-2. The review also ignores all other syncytins except syncytin-1, which is only involved in human and closely-related primate placentation and not involved in placentation of most mammals. This is not consistent with the objective of the review to discuss “exaptation of retroviral syncytin for development of placental mammals”.

Response: Although we agree with the reviewer that the cited above descriptions of various types of placenta do not have a direct relation to the syncytin issues, we would like to keep these descriptions because they are very rarely covered in the literature and are very important for the understanding of the multi-branched evolution of placental mammals. We also, as mentioned above, changed the title and the introduction to more accurately describe the article premise.

Other syncytins are only mentioned when convenient (for instance, in the last sentence of the immunomodulatory paragraph where syncytin-2 is discussed).

Response: the reviewer is incorrect, we described other syncytins in the paragraph at the end of the “Endogenous retroviral gene function in the placenta” subchapter. But because the reviewer missed it, to make it more pronounced we separated this section under the title “Other syncytins” and expanded it by adding more information on the other types of syncytins

 More specific comments for improvement are provided:    

Major comments:

  • The introduction and first few sections have generalizations which are not accurate, such as:
    • “In the simplest term, the mammalian placenta is a temporary fetal/maternal organ formed through the tight apposition of the specialized fetal syncytial tissue with the wall of the uterus.” This is an incorrect generalization. As an example, in human placenta, extravillous trophoblasts ultimately form the major interface with the decidua (endometrial stroma, not wall of the uterus). Other species have drastically different types of placentation, only some of which appose the wall of the uterus and few of these involve syncytin-generated syncytia.

 Response: As requested, we corrected and changed these statements and    descriptions in Abstract and Introduction

    • “The syncytialization process is essential for placentation because only the syncytial trophoblast has the invasive capacity allowing intimate contact with the uterine endometrium and the formation of the placenta. Thus, the cell fusion occurring during syncytialization plays a pivotal role in placenta formation, and consequently, in the development and reproduction of placental mammals.” This is true in human placentation during the first few days of implantation, but at later stages mononuclear “extravillous” trophoblasts comprise the primary invading trophoblasts. An invading syncytialized trophoblast is also not a conserved feature of placentation in mammals.

Response: we changed these descriptions in the Introduction and the subchapter on the types of placenta

    • About half the introduction is a description of nonmammalian placentas. What is the point of including these in the introduction? Do these placenta-like structures undergo cell fusion? Do they express syncytins?

Response: as we responded above, the cited above descriptions of various types of placenta do not have a direct relation to the syncytin issues, we would like to keep these descriptions because they are very rarely covered in the literature and are very important for the understanding of the multi-branched evolution of placental mammals. We also added information on cell fusion and syncytin expression in these placentas and added the references.

  • There is a large section on the role of syncytin-1 in fertilization. To date, there has only been one report that synctyin-1 and its receptor is expressed in human gametes, and no functional studies (the current review includes two articles, but one is a review article). There is also no evidence that syncytin-related genes are involved in fertilization in any other species. For example, Syncytin-A and Syncytin-B deficient mouse embryos implant and are viable up until at least mid-gestation. Thus, the notion that syncytins are absolutely required for different phases of egg-sperm fusion (as suggested in Figure 3), and that they may have instigated fertilization in all eukaryotes, should be removed altogether or relegated to a sentence or two speculating that syncytins could be involved in fertilization in some species but functional studies are required.

Response: We would like to keep this subchapter because one of the worries about the vaccine is that it will affect sperm and fertilization, but we changed this subchapter and Figure 3 to be more hypothetical. Also, the reviewer is not fully correct about the mouse syncytin A-B. There are experimental data indicating that that these syncytins are required for the formation of the two-layered murine placental syncytiotrophoblast, and syncytin-A knockout mice demonstrate its critical role in placentation (Dupressoir et al., 2009, 2011). As requested in the other comment above, we added the subchapter on the different syncytins and their roles.

It is surprising that syncytiotrophoblast is emphasized as an invasive structure, which is rare in most mammals, and it is not mentioned for its primary function as lining the exchange surface regulating transfer of substances between maternal and fetal blood in some placentae (humans and a few other mammals), or for its critical role in hormone production and metabolism.

Response: As suggested we rephrased this section

  • The section on the syncytin-1 comparison to COV-2 reads more like an opinion piece than a scientific report, with language that includes “misinformed skeptics”, and “erroneous claims”.

Response: we softened/removed these terms

  • Given that more than half the title is devoted to homology between syncytin-1 and Spike protein, it was surprising that the article only contains a BLAST search comparing amino acid sequences and a single paragraph rebuking skeptics. For instance, there is no discussion on the similar genetic elements in which many of the concerns of similarity are based, three-dimensional structure of each protein and their different affinities for unique receptors, or how there are differences between CoV and CoV2 spike proteins, the former having previously identified similarities with Syncytin-1.

Response: Although we agree with the reviewer comment, we do not want to add this information for the following reasons: the main goal of our review/commentary was to refute the arguments of the anti-vaccination proponents. We wanted to present scientifically sound but easy-to-understand explanations why the COVID-19 vaccine is most likely safe and will not cause infertility or placentation problems. We think that adding the more detailed and complex information on the three-dimensional and genomic structure etc. would somehow “muddle” our message. Instead, as requested below, we added available information on COVID-19 vaccine safety.

In line with the above statement, the conclusions of this review article are based on opinion: “Thus, in summary, the very limited similarity between the SARS-CoV-2 spike protein and human syncytin-1 assures that the COVID-19 vaccines are very specific, and they will not recognize (or destroy) syncytin protein, and will not affect, fertility, pregnancy, or cause cancer.” While this is certainly hopeful, there is no way for the authors to be able to make these claims based only on a BLAST search comparing the CoV-2 spike protein and syncytin-1. It would be more accurate to assert that based on the limited sequence similarity between these proteins, it is unlikely that any vaccine targeted against the CoV-2 spike protein would generate an immune response against syncytin-1.

Response: as suggested, we changed the indicated sentences/paragraph

  • The review could also benefit from including some sentences about the current evidence suggesting that those who have had COVID19 or have received vaccines are fertile, able to conceive, and carry a pregnancy, which indicates that the vaccine (or virus) is not likely to impact these process in the short-term. However, it is prudent to continue to evaluate any potential impact of COVID19 and/or vaccines on fertility and health.

Response: as suggested, we added the information of the safety of COVID-19 vaccine, and also added the statement about the importance of monitoring for any potential impact of the vaccine on these processes and added the references.

Minor comments:

  • “Have four main types (with a variety of subtypes) of the placenta”…there are different ways to classify placentas, of which the degree of layers separating maternal blood and trophoblast is one strategy.

Response: we changed this statement

  • “Although it would seem that the noninvasive placenta, as potentially the least effective in nourishing the fetus because of the lack of direct contact with maternal blood, is evolutionary most ancient, and thus primitive, this is not the case.” This is a hypothesis based on some excellent comparative placentation studies, so the wording should be changed to reflect that “this does not appear to be the case” or “this is not likely the case”.

Response: We rephrased this statement

  • “Thus, the COVID-19 vaccines are targeting the viral spike protein.” – is there a reference for this statement? Maybe soften this statement to enable some flexibility for different vaccine strategies, such as “the viral spike protein is the primary target antigen for many vaccine manufacturers”

Response: We rephrased this statement

  • The review is not comprehensive, consisting of just over 60 references, and requires significant editing. “The” is overused when describing terms, for example (page 7): “The fusion is the local rupture…” and “…when there the fusion is different…”. There are also many awkward sentences. As one example, in Section 3: “Although during the evolution, the majority of TEs rearranged and lost their original functions/coding capacity, the evolutionarily youngest or/and domesticated for the fulfillment of the critical for the host organism functions are still functional. An example of such necessary for the host retroviral proteins are retroviral env genes”.

Response: We fully agree that this is not a comprehensive review, however, our goal never was to write a comprehensive review, but to comment on the main facts and arguments used by the anti-vaccination movement to scare the public. As suggested, we also re-edited the text and rewrote the awkward sentences.

Round 2

Reviewer 1 Report

The authors have addressed all my concerns.

Author Response

We thank the referee for her/his constructive comments.

Reviewer 2 Report

The authors have adequately responded to the previous concerns.

A quick editing note just for clarity: in the reviewer comments it was stated that "there is also no evidence that syncytin-related genes are involved in fertilization in any other species. For example, Syncytin-A and Syncytin-B deficient mouse embryos implant and are viable up until at least mid-gestation." The authors responded stating that this statement was incorrect because syncytin A and B are required for syncytiotrophoblast formation and placentation. This reviewer is fully aware that syncytin A and B are required for mouse placentation. However, sperm and egg still fuse in mice lacking syncytin A and B. Hence, these syncytins are likely not required for fertilization (sperm-egg fusion) because viable conceptuses implant, and have a viable pregnancy at least until midgestation (in the case of syncytin A-knockouts) or term (syncytin B-knockouts). This evidence makes it very unlikely that syncytin A or syncytin B are involved in sperm-egg fusion.  

Author Response

Reviewer 2:

A quick editing note just for clarity: in the reviewer comments it was stated that "there is also no evidence that syncytin-related genes are involved in fertilization in any other species. For example, Syncytin-A and Syncytin-B deficient mouse embryos implant and are viable up until at least mid-gestation." The authors responded stating that this statement was incorrect because syncytin A and B are required for syncytiotrophoblast formation and placentation. This reviewer is fully aware that syncytin A and B are required for mouse placentation. However, sperm and egg still fuse in mice lacking syncytin A and B. Hence, these syncytins are likely not required for fertilization (sperm-egg fusion) because viable conceptuses implant, and have a viable pregnancy at least until midgestation (in the case of syncytin A-knockouts) or term (syncytin B-knockouts). This evidence makes it very unlikely that syncytin A or syncytin B are involved in sperm-egg fusion.  

RESPONSE: We are very sorry for misunderstanding and we fully agree with the reviewer. For further clarity we added the information that it is very unlikely that syncytin A and B are involved in fertilization. We also previously added the question marks to the Figure 3 about the potential involvement of syncytin in fertilization.

In addition we slightly modified the title and added the following in the last sentence: Although the further evaluation of any potential impact of COVID19 and/or vaccines on fertility, pregnancy, placentation, and general health is necessary

All changes are highlighted in red in the second revision.
